# Mutation of Basic Residues R283, R286, and K288 in the Matrix Protein of Newcastle Disease Virus Attenuates Viral Replication and Pathogenicity

**DOI:** 10.3390/ijms24020980

**Published:** 2023-01-04

**Authors:** Zhiqiang Duan, Haiying Shi, Jingru Xing, Qianyong Zhang, Menglan Liu

**Affiliations:** 1Key Laboratory of Animal Genetics, Breeding and Reproduction in the Plateau Mountainous Region, Ministry of Education, Guizhou University, Guiyang 550025, China; 2College of Animal Science, Guizhou University, Guiyang 550025, China

**Keywords:** Newcastle disease virus, matrix protein, basic residue, viral replication

## Abstract

The matrix (M) protein of Newcastle disease virus (NDV) contains large numbers of unevenly distributed basic residues, but the precise function of most basic residues in the M protein remains enigmatic. We previously demonstrated that the C-terminus (aa 264–313) of M protein interacted with the extra-terminal (ET) domain of chicken bromodomain-containing protein 2 (chBRD2), which promoted NDV replication by downregulating chBRD2 expression and facilitating viral RNA synthesis and transcription. However, the key amino acid sites determining M’s interaction with chBRD2/ET and their roles in the replication and pathogenicity of NDV are not known. In this study, three basic residues—R283, R286, and K288—in the NDV M protein were verified to be responsible for its interaction with chBRD2/ET. In addition, mutation of these basic residues (R283A/R286A/K288A) in the M protein changed its electrostatic pattern and abrogated the decreased expression of endogenic chBRD2. Moreover, a recombinant virus harboring these mutations resulted in a pathotype change of NDV and attenuated viral replication and pathogenicity in chickens due to the decreased viral RNA synthesis and transcription. Our findings therefore provide a better understanding of the crucial biological functions of M’s basic residues and also aid in understanding the poorly understood pathogenesis of NDV.

## 1. Introduction

Newcastle disease virus (NDV) is an important avian pathogen that usually causes a highly contagious disease in chickens, resulting in considerable economic losses to the poultry industry worldwide [1]. The NDV has a non-segmented, negative-sense, single-stranded RNA genome of approximately 15.2 kb in length, which encodes six structural proteins (nucleocapsid (NP) protein, phosphoprotein (P) protein, matrix (M) protein, fusion (F) protein, hemagglutinin-neuraminidase (HN) protein, and large polymerase (L) protein) and two non-structural proteins (V and W) [2,3]. Among the above viral proteins, the M protein is surrounded by the inner surface of the viral envelope and is involved in the morphogenesis of NDV [4]. It has been demonstrated that the NDV M protein is a multifunctional nucleocytoplasmic trafficking protein. Besides the participation of assembly and budding of progeny virions in the cytoplasm late in infection [5], the M protein is proved to inhibit host cell transcription and translation during its early nuclear localization [6,7]. In addition, an increasing number of studies have shown that some amino acid motifs (such as ^23^FPIV^26^, ^247^KKGKKVTFDKLEERKIRR^263^, and ^274^LGP^276^) or sites (such as R36, R42, and K247) in the M protein are associated with the virulence and replication of NDV [8,9,10,11,12,13]. Moreover, the interactions between M protein and cellular proteins are also important for the replication and pathogenicity of NDV [9,14,15,16,17]. Therefore, all of these findings give evidence of the pivotal roles of M protein in the life cycle of NDV.

The M protein of most paramyxoviruses has highly basic and hydrophobic characteristics, of which nuclear localization signal and nuclear export signal composed of multiple basic residues and hydrophobic residues determine the nucleocytoplasmic shuttling of M protein [18]. As an important member of the paramyxoviruses, the NDV M protein also has highly basic, hydrophobic, but not membrane-spanning properties [19], which are consistent with the known peripheral attachment of the M protein to the viral envelope [4]. However, the basic residues are not distributed uniformly throughout the sequences of M protein. Meanwhile, the precise functions of these basic residues in the NDV M protein remains enigmatic. To date, only a few basic residues in the M protein have been investigated. For example, the R36 residue is identified as a unique amino acid and positive-selection site in the M protein of pigeon-origin NDV, which enhances the adaptation and pathogenicity of NDV in pigeons [11]. Beyond that, a basic residue R42 and two basic amino acid clusters (^247^KKGKKVTFDKLEERKIRR^263^) in the NDV M protein assist its nuclear localization and promote viral replication and pathogenicity [9,12]. Recently, the K247 residue in the M protein was verified to be an important monoubiquitination site, which regulates the nuclear-cytoplasmic trafficking of M protein and enhances the virulence and replication of NDV [13]. These results clearly indicate that the basic residues of M protein play important roles in the replication and pathogenicity of NDV. However, more studies are needed to explore the role of other basic residues of M protein in the regulation of NDV replication in detail.

Bromodomain-containing protein 2 (BRD2) belongs to the bromodomain and extra-terminal domain family, which plays crucial roles in chromatin remodeling, cell cycle progression, inflammatory responses, etc. [20]. In recent years, BRD2 has also been found to participate in the replication of multiple DNA or RNA viruses via interacting with viral proteins [21,22,23,24]. In our previous studies, we demonstrated that chicken BRD2 (chBRD2) was a novel interacting partner of NDV M protein [9], and the C-terminus (aa 264–313) of M protein was responsible for direct interaction with the extra-terminal (ET) domain (aa 619–683) of chBRD2 [17]. However, which key residues of M protein interact with chBRD2 and how these residues regulate the replication and pathogenicity of NDV remain unknown. In this study, three basic residues—R283, R286, and K288—in the NDV M protein were verified to determine its interaction with chBRD2/ET. We further demonstrated that M protein harboring R283A/R286A/K288A mutations not only changed its electrostatic pattern and abrogated the decreased expression of endogenic chBRD2, but also attenuated the pathotype, replication ability, and pathogenicity of recombinant NDV due to the decreased viral RNA synthesis and transcription. These results provide valuable information for better understanding the biological functions of the basic residues in NDV M protein, with important implications for the replication and pathogenesis of NDV.

## 2. Results

### 2.1. Basic Residues R283, R286, and K288 of M Protein Are Crucial for Its Interaction with chBRD2/ET

The C-terminus (aa 264–313) of NDV M protein and the ET domain of chBRD2 have been demonstrated to be responsible for interaction with each other (Figure 1A) [17]. To further verify the key amino acid sites of M protein interacting with chBRD2/ET, the regions (aa 261–313) of M protein were first divided into different truncation mutants, and then used to examine their interactions with chBRD2/ET by co-immunoprecipitation (Co-IP) assay. The results showed that the Myc-chBRD2/ET could be immunoprecipitated with HA-M (264–313), HA-M (264–292), and HA-M (281–288) proteins rather than HA-M (289–313) and HA-M (264–280) proteins (Figure 1B). In turn, only the HA-M (264–313), HA-M (264–292), and HA-M (281–288) proteins could be immunoprecipitated with chBRD2/ET (Figure 1B). Next, the amino acid sites within aa 281–288 of M protein were separately mutated into alanine (A) to detect the interactions between M (264–313) and chBRD2/ET. It was interesting that R283A, R286A, or K288A single mutation reduced the binding ability of HA-M (264–313) to Myc-chBRD2/ET, and R283/R286A/K288A (3A) mutations absolutely abrogated their interactions (Figure 1C), thus indicating that the basic residues R283, R286, and K288 of NDV M protein determined its interaction with chBRD2/ET.

### 2.2. The M/3A Protein Has a Changed Electrostatic Pattern

To investigate whether the M protein harboring 3A mutations changes its three-dimensional (3D) structure, 3D structural models of the M and M/3A proteins were constructed and compared. The results of secondary structure composition analysis revealed that the secondary structure of M protein was mainly composed of α-helices (from α1 to α13) and β-sheets (from β1 to β14) (Figure 2A), which was consistent with the previous findings (4). The three basic residues R283, R286, and K288 were distributed between the β11 and α10 (Figure 2A). However, there were no obvious changes in the 3D structural model between M and M/3A proteins (Figure 2B). Meanwhile, the secondary structure in the cartoon drawing of the monomeric structure of the M and M/3A proteins was also not changed (Figure 2C). Further in-depth analysis indicated the surface charge surrounding the mutated amino acids of the M/3A protein was altered when compared with the M protein (Figure 2D, marked with a green circular dotted line). Therefore, these results suggested that the change in the electrostatic pattern caused by 3A mutations in the M protein was responsible for the disruption of the M-chBRD2 interaction.

### 2.3. The M/3A Protein Abrogates the Decreased Expression of Endogenic chBRD2

The NDV M protein has been proven to negatively regulate the expression of chBRD2 in a dose-dependent manner [17]. To determine whether the M protein harboring 3A mutations affects its subcellular localization and the expression of endogenic chBRD2, experiments with DF-1 cells transfected with pCMV-HA, pCMV-HA-M, and pCMV-HA-M/3A were performed. As shown in Figure 3A, the HA tag showed an obviously cytoplasmic localization, while the recombinant proteins HA-M and HA-M/3A exhibited a predominantly nuclear localization. In addition, the expression levels among the HA tag, HA-M, and HA-M/3A proteins had no significant difference (Figure 3B). However, the expression of recombinant protein HA-M rather than HA tag or HA-M/3A protein significantly reduced the expression of chBRD2 (*p* < 0.01) (Figure 3C). Thus, these results suggested that the M protein harboring 3A mutations lost its ability in reducing the expression of endogenic chBRD2.

### 2.4. The M/3A Protein Changes the Biological Characteristics of Recombinant NDV

Now that 3A mutations in the M protein had affected its surface charge, we further investigated the effect of M/3A on the biological characteristics of NDV. The recombinant NDV harboring M/3A mutations was rescued by reverse genetics approaches, and then identified by indirect immunofluorescence assay (IFA) using chicken anti-NDV serum. The results showed that green fluorescence was obviously detected in DF-1 cells infected with rSS1 and rSS1-M/3A (see Appendix A), demonstrating that the recombinant virus was successfully rescued. To determine the stability of the mutant virus, the rescued virus rSS1-M/3A was plaque purified and passaged five times in 10-day-old specific pathogen free (SPF) embryonated chicken eggs. Sequence analysis of the whole genome of rSS1-M/3A after five passages showed that the introduced 3A mutations in the M protein were unaltered (see Appendix A), and no additional mutations were observed in the *M* gene or other viral genes. In addition, the biological characteristics detection of these viruses revealed that 3A mutations in the M protein significantly extended the MDT of rSS1-M/3A (102 ± 2 h) in embryonated chicken eggs compared with rSS1 (53 ± 2 h), and the ICPI value of rSS1-M/3A (1.63 ± 0.01) was lower than that of rSS1 (1.90 ± 0.02) (Table 1). Meanwhile, consistent with the results of MDT and ICPI, the growth ability (measured as EID_50_ and TCID_50_) of rSS1-M/3A in embryonated eggs and cells also showed a significant decrease (Table 1).

### 2.5. The M/3A Protein Causes the Reduced Viral RNA Synthesis and Transcription of Recombinant NDV

Because the interaction of M protein with chBRD2 can downregulate chBRD2 expression to facilitate viral RNA (vRNA) synthesis and transcription [17], we speculated that recombinant NDV expressing the M/3A protein will affect vRNA synthesis and transcription. Therefore, the vRNA levels and mRNA levels of viral NP and P genes in rSS1- and rSS1-M/3A-infected cells were analyzed. As shown in Figure 4A, the relative levels of vRNA (corresponding to the *NP* and *P* genes) in rSS1-M/3A-infected cells were more decreased than those in rSS1-infected cells at 12 hpi (*p* < 0.01). Meanwhile, there were also statistical differences in the relative vRNA levels between cells infected with rSS1 and rSS1-M/3A at 18 hpi (*p* < 0.05). Similarly, we found that the relative mRNA levels of NP and P genes in rSS1-M/3A-infected cells were also much lower than those in rSS1-infected cells at 6 hpi (*p* < 0.05), and significantly lower at 12 and 18 hpi (*p* < 0.01) (Figure 4B). In addition, the expression levels of viral NP and HN proteins were also relatively decreased during the course of rSS1-M/3A infection at 6, 12, and 18 hpi (Figure 4C), which was consistent with the results of viral mRNA expression levels. Collectively, these results indicated that 3A mutations in the M protein could reduce vRNA synthesis and transcription due to the abrogated inhibition of chBRD2 expression.

### 2.6. The M/3A Protein Leads to Decreased Replication Ability of Recombinant NDV

To further compare the replication kinetics of rSS1 and rSS1-M/3A, the multicycle growth kinetics of these viruses in DF-1 cells were examined. As shown in Figure 5A, the virus titers of rSS1-M/3A were remarkably reduced in comparison to that of rSS1 from 12 to 96 h post-infection (hpi) (*p* < 0.01). In addition, the effect of 3A mutations on the subcellular localization of M protein was evaluated. The results of IFA showed that the M protein of rSS1 and rSS1M/3A was primarily localized in the nucleus at 6 hpi and gradually concentrated in the nucleus at 12 hpi, and then mainly observed in the cytoplasm at 18 hpi (Figure 5B). Although there was no obvious difference in the subcellular localization of rSS1 and rSS1-M/3A M proteins, the expression level of rSS1-M/3A M protein in the nucleus and/or cytoplasm was much lower than that of rSS1 protein at 6, 12, and 18 hpi (*p* < 0.01), respectively (Figure 5C). Moreover, rSS1-M/3A harboring the M/3A protein did not reduce the expression of endogenic chBRD2 in comparison to the rSS1 infection group at different time points (Figure 5D), which was coincident with the results of plasmid-transfected cells (Figure 3C). Together, these data indicated that 3A mutations in the M protein reduced the replication ability of recombinant NDV.

### 2.7. The M/3A Protein Attenuates the Pathogenicity of Recombinant NDV in SPF Chickens

To further evaluate the effect of M/3A on the pathogenicity of NDV, the in vivo pathogenesis assessment of rSS1-M/3A in 4-week-old SPF chickens was then executed. The resulting survival curves showed that birds inoculated with the parental virus rSS1 exhibited slight depression at 3 days post-infection (dpi), severe depression (3/10), wing drop (2/10), hemiparesis/paralysis (3/10), and death (2/10) at 4 dpi, and 100% mortality by 5 dpi (Figure 6A). At necropsy, all euthanized chickens infected with rSS1 presented obvious and severe gross lesions in the organs, such as conjunctivitis, hemorrhage of the trachea, thymus, throat, proventriculus, and duodenum mucosa, accompanied by multifocal necrosis in the spleen, and marked atrophy of the thymus and bursa of Fabricius at 4–5 dpi. On the contrary, birds infected with rSS1-M/3A showed a delayed and slight depression (5/10) at 4 dpi, severe depression (1/10) and two deaths (1/10) at 5 dpi, and then four deaths (2/10) at 6 dpi, while there were no deaths in the subsequent days (Figure 6A). The results of the virus titration assay revealed that rSS1-M/3A exhibited little replication ability in the spleen, lung, brain, thymus, bursa of Fabricius, and duodenum at 4 dpi; by contrast, rSS1 could replicate in multiple tissues and had relatively higher virus titers in the lymphoid tissues (spleen, thymus, and bursa of Fabricius) than rSS1-M/3A (Figure 6B).

In addition, the results of tissue histopathology observation showed that rSS1-M/3A and rSS1 caused the following moderate to several histopathological changes in the tissues of the spleen, thymus, bursa of Fabricius, lung, brain, and duodenum at 4 dpi, respectively (Figure 6C): birds infected with rSS1 displayed multifocal confluent coagulative necrosis, severe lymphocyte depletion, and infiltration of macrophages in the lymphoid tissues (spleen, thymus, and bursa of Fabricius) (empty arrows); extreme expansion in the bronchi of the lung (empty arrows); loose brain tissue structure with increased accumulation of microglial cells in the brain (empty arrows); and mucosal epithelium shedding and inflammatory cell infiltration of the mucosal lamina propria in the duodenum (empty arrows). However, only slight histopathological changes were observed in the above tissues from the rSS1-M/3A group (black arrows), and as expected, no apparent histopathological changes were found in any of the above tissues from the control group inoculated with PBS (Figure 6C). Together, these results demonstrated that 3A mutations in the M protein could attenuate the replication and pathogenicity of NDV in SPF chickens.

## 3. Discussion

The basic residues in the M protein of most paramyxoviruses usually function as NLS determining its nuclear transport [25,26,27]. However, several studies have demonstrated that the basic residues of some paramyxovirus M proteins possess additional functions. For example, a highly conserved basic residue K258 in the M protein of Nipah virus and Hendra virus, and K247 in the M protein of NDV, are important monoubiquitination sites which regulate the nuclear-cytoplasmic trafficking of M protein and viral budding [13,28,29]. In addition, the K258 residue of Nipah virus M protein is also responsible for targeting the E3-ubiquitin ligase TRIM6 to inhibit the IKKε kinase-mediated type-I interferon antiviral response [30]. In addition, a recent study reported that several lysines clustering in a basic patch microdomain of the C-terminal domain of canine distemper virus M protein contribute to plasma membrane association and viral budding activity [31]. Up until now, only a small number of basic residues, such as R36, R42, K247, and two basic amino acid clusters of NLS in the NDV M protein have been investigated [9,11,12,13]. Given the highly basic characteristics of NDV M protein, more efforts are needed to elucidate the crucial roles of M‘s basic residues in the life cycle of NDV.

It has been shown that mutation of important functional amino acid sites in the NDV M protein can change its partial secondary structure and tertiary structure [9,10,13]. According to the previously reported interactions of the C-terminus (aa 264–313) of NDV M protein with chBRD2/ET, in this study, three basic residues (R283, R286, and K288) in the NDV M protein were verified to be the key sites determining its interaction with chBRD2/ET. Interestingly, the results of structural model analysis generated by AlphaFold2 and PyMoL revealed that both the 3D structural model and the secondary structure of the M and M/3A proteins were not changed, but the M/3A protein showed the changed surface charge surrounding the mutated amino acids, which is different from the previous studies. These results indicated that the three basic residues played a crucial role in supporting the electrostatic pattern of NDV M protein, which determines the accurate interaction between M protein and chBRD2.

BRD2 is reported to interact with many chromatin- and transcriptional activation-related proteins to regulate the transcription of diverse genes [32,33,34]. Importantly, several studies have shown that viruses hijacking BRD2 to enhance viral gene transcription is beneficial for viral replication [22,23,35,36]. Consistent with these findings, we previously found that the expression level of chBRD2 was obviously reduced in both NDV-infected cells and pEGFP-M-transfected cells, which facilitated vRNA synthesis and transcription [17]. However, in this study, the M protein harboring R283A/R286A/K288A mutations abrogated the decreased expression of endogenic chBRD2 in pCMV-HA-M/3A-transfected cells and recombinant NDV-infected cells, along with the decreased viral RNA synthesis and transcription. It was worth noting that although the subcellular localization of M/3A protein was not altered, the expression levels of viral M, NP, and HN proteins were reduced. We speculated that the M/3A protein lost its inhibitory effect in decreasing chBRD2 expression, which reduced the expression of viral ribonucleoprotein and in turn affected vRNA synthesis and transcription to attenuate the pathotype and replication of NDV. Because BRD2, together with TBP-associated factors (TAFs), chromatin/histone modification enzymes, activated transcription factors, RNA Polymerase II, and SWI/SNF remodeling complex components, constitute the major interaction subcomplexes to regulate gene transcription [32], whether the M/3A protein reduces the expression of BRD2-involved transcription complexes and how the M/3A protein plays transcriptional inhibition to regulate viral ribonucleoprotein synthesis remain to be investigated.

In recent years, increasing lines of evidence have indicated that BRD2 plays crucial roles in regulating NF-κB activity [37,38,39,40] and interferon production [41,42]. The NDV M protein is demonstrated to have an innate immune antagonism feature, which inhibits inflammatory responses through TIFA/TRAF6/NF-κB or IRAK4/TRAF6/TAK1/NF-κB signaling pathways [7,43]. Meanwhile, the reduced production of inflammatory cytokines caused by the M protein promotes the replication and cytopathogenicity of NDV. Here, we found that the recombinant NDV harboring M/3A mutations showed the non-virulent (lentogenic) pathotype compared with the highly virulent (velogenic) pathotype of the parent virus rSS1. In addition, the virus titers of rSS1-M/3A in cells and eggs were obviously decreased, and the replication ability and tissue pathological changes of rSS1-M/3A in SPF chickens were also greatly attenuated. We concluded that the 3A mutations disrupted the interaction of M protein with chBRD2, which might enhance BRD2-mediated inflammatory responses to resist NDV infection. Therefore, further studies will be performed to identify the relationship among M/chBRD2/NF-κB and to elucidate in-depth how the M/3A protein affects chBRD2 expression to regulate the replication and pathogenicity of NDV.

## 4. Materials and Methods

### 4.1. Cell line, Virus, and Antibodies

Chicken embryonic fibroblast cell line (DF-1) and baby hamster kidney cell line (BHK-21) stably expressing T7 RNA polymerase (BSR-T7/5) were kind gifts from Prof. Xiufan Liu (Yangzhou University, Yangzhou, China). The virulent NDV strain SS1 (GenBank no. KP742770.1) was isolated and preserved in our laboratory [44]. The rabbit anti-M, NP, and HN polyclonal antibodies were prepared by Wuhan GeneCreate Biological Engineering Co., Ltd. (Wuhan, China). The rabbit polyclonal antibodies against chBRD2 (DF12857) and GAPDH (AF7021) and the mouse monoclonal antibodies against HA tag (T0008) and Myc tag (T0001) were purchased from Affinity Biosciences (Cincinnati, OH, USA).

### 4.2. Animals and Ethics Statement

SPF embryonated eggs (10-day-old) and SPF chickens (1-day-old or 4-week-old) were purchased from Beijing Boehringer Ingelheim Vital Biotechnology Co., Ltd. (Beijing, China). The animal experimental protocol was approved by Experimental Animal Ethics of Guizhou University (No. EAE-GZU-2022-T007).

### 4.3. Plasmid Construction

The recombinant plasmids expressing the full-length M protein (pCMV-HA-M), the truncation mutant of M protein pCMV-HA-M (264–313), and the ET domain of chBRD2 (pCMV-Myc-chBRD2/ET) were generated in our previous study [17]. As for the construction of M’s truncation mutants, annealed oligonucleotides encoding the fragments, including M (264–292), M (289–313), M (264–280), M (281–288), M/mt1, M/mt2, M/mt3, M/mt4, M/mt5, M/mt6, and M/mt7, were inserted into pCMV-HA (an empty plasmid expressing the influenza haemagglutinin (HA), which was purchased from Clontech (San Francisco, CA, USA)) to generate the corresponding recombinant plasmids. In addition, overlapping PCR was performed to construct the recombinant plasmid pCMV-HA-M/3A (HA-tagged M protein harboring R283A/R286A/K288A mutations).

To generate a recombinant cDNA clone containing R283A/R286A/K288A substitutions within the M protein, the fragment carrying the restriction enzyme sites AgeI and BstZ17I was amplified by two pairs of specific primers to produce two overlapping PCR fragments. The two PCR fragments were joined in a second round of PCR and the obtained fragment was digested with *Age* I and *Bst*Z17 I to replace the corresponding region in the full-length cDNA clone pNDV/SS1 [45] (see Appendix A). The resulting plasmid was named pNDV/SS1-M/3A. Primers used for the construction of the above recombinant plasmids are shown in Appendix A. All of the constructed recombinant plasmids were jointly verified by PCR, restriction digestion, and DNA sequencing.

### 4.4. Sequence Analysis and Molecular Modelling of M Protein

The amino acid sequences of NDV M protein (GenBank no. AJM13618.1) were analyzed by A Protein Secondary Structure Prediction Server (https://www.compbio.dundee.ac.uk/www-jpred/, accessed on 25 August 2022) to understand its secondary structure composition. In addition, the protein structural models of M and M/3A were generated by AlphaFold2 (John Jumper, DeepMind, London, UK). The 4G1G (https://www.rcsb.org/structure/4g1g, accessed on 22 December 2022) crystal structure of Newcastle disease virus (STRAIN AUSTRALIA-VICTORIA/32) M protein is solely the structural template. All of the structural annotations were generated using PyMOL software (version 2.5.4, Schrödinger, San Diego, CA, USA).

### 4.5. Cell Culture and Plasmid Transfection

DF-1 cells were cultured in Dulbecco’s modified Eagle medium (DMEM) (Gibco, New York, NY, USA) containing 10% fetal bovine serum (FBS) (Gibco, New York, NY, USA) at 37 °C under an atmosphere with 5% CO_2_. For the plasmid transfection experiments, 3 × 10^5^ DF-1 cells were grown to 80% confluence in 35 mm dishes and then double- (the plasmid expressing M truncation mutants and pCMV-Myc-chBRD2/ET) or single- (pCMV-HA, pCMV-HA-M, or pCMV-HA-M/3A) transfected with a total of 3 μg of plasmid using FuGENE HD Transfection Reagent (Promega, Madison, WI, USA) according to the manufacturer’s instructions. Twenty-four hours after transfection, DF-1 cells expressing the recombinant proteins were used for the subsequent Co-IP or IFA experiment.

### 4.6. Co-IP Assay

For Co-IP assay, 3 × 10^5^ DF-1 cells cultured in 35 mm dishes were co-transfected with the indicated plasmids co-expressing the truncation mutants of M protein and chBRD2/ET. At 24 h post-transfection (hpt), cells were washed thrice with phosphate-buffered saline (PBS) and then lysed with immunoprecipitation lysis buffer (Pierce, TX, USA). The supernatants were collected and then incubated with anti-HA or anti-Myc antibodies overnight at 4 °C. The immune complexes were adsorbed by protein A+G agaroses (Sigma-Aldrich, St. Louis, MO, USA) for 2 h at room temperature. After three washes with immunoprecipitation lysis buffer, the immunoprecipitates were detected by Western blotting using anti-Myc or anti-HA antibodies.

### 4.7. IFA

DF-1 cells cultured in 35 mm dishes were transfected with the plasmids expressing HA tag, HA-M, or HA-M/3A protein. At 24 hpt, cells were rinsed thrice with PBS, fixed with 4% paraformaldehyde for 15 min, and then permeabilized with 0.25% Triton X-100 for 5 min. Cells were rinsed thrice with PBS and blocked with 10% FBS in PBS for 30 min at 37 °C, and then incubated with mouse anti-HA antibody diluted in PBS containing 10% FBS for 1 h at 37 °C. After three washes with PBS, the cells were incubated with FITC-labeled goat anti-mouse IgG(H+L) antibody (Beyotime Biotechnology, Shanghai, China) for 1 h at 37 °C. Cells were counterstained with DAPI (Sigma, USA) to detect the nuclei. Images were captured with an inverted fluorescence microscope and then dealt with Adobe Photoshop 8.0 software.

### 4.8. Recombinant Virus Rescue and Virus Infection

Rescue of the recombinant virus was performed as previously described (9). Briefly, BSR-T7/5 cells were grown in DMEM medium containing 10% FBS and 1 mg/mL Geneticin G418 with five consecutive passages before transfection. Cells at 80% confluence in 35 mm dishes were transfected with the pNDV/SS1-M/3A together with three SS1-derived helper plasmids (pCI-NP, pCI-P, and pCI-L) using FuGENE HD transfection reagent (Promega, USA). At 72 hpt, cells were harvested and inoculated into the allantoic cavities of 10-day-old SPF embryonated eggs. The hemagglutination (HA) test, IFA, and DNA sequencing were used to identify the rescued virus rSS1-M/3A. The pathogenicity assay of the rescued virus was determined using the standard tests [46].

For virus infection experiment, low-passage DF-1 cells grown to 80% confluence in 6-well plates were infected with rSS1 and rSS1-M/3A at an MOI of 0.1. The cell culture supernatants were collected at the indicated time points (6, 12, 24, 36, 48, 72, and 96 hpi), and the virus titers were titrated using 50% tissue culture infective doses (TCID_50_) in DF-1 cells according to the Reed and Muench method [47]. In addition, the subcellular localization of rSS1 and rSS1-M/3A M proteins was observed at 6, 12, and 18 hpi using IFA as described above. Meanwhile, the expression of the M, NP, HN, and chBRD2 proteins in rSS1- and rSS1-M/3A-infected cells was examined by Western blotting at 6, 12, and 18 hpi, respectively. The relative levels of the indicated proteins compared with control GAPDH or Histone H3 expression were determined by densitometry using ImageJ software (version 1.8.0, National Institutes of Health, New York, NY, USA).

### 4.9. Quantification of vRNA Synthesis and Transcription

DF-1 cells infected with rSS1 and rSS1-M/3A were collected at 6, 12, and 18 hpi, respectively, and total RNA was extracted and reverse-transcribed using a previously reported method [48]. Primer sequences used for the quantification of viral RNA synthesis and transcription were derived from previous reports [49]. Quantitative real-time PCR (qRT-PCR) experiments were performed using a SYBR Premix Ex Taq Kit (Takara Biomedical Technology, Dalian, China). The qRT-PCR procedure used to quantify vRNA synthesis and transcription was carried out as described previously [48]. The comparative threshold cycle (ΔΔC_T_) method was used to calculate the relative abundance of mRNA and vRNA. The relative gene expression levels were normalized to those of the *GAPDH* gene.

### 4.10. Pathogenicity in 4-Week-Old SPF Chickens

The pathogenesis assessment of rSS1 and rSS1-M/3A was determined in 4-week-old SPF chickens. Thirty-nine chickens were assigned randomly into three experimental groups, including rSS1- (n = 13), rSS1-M/3A- (n = 13), and mock- (PBS, n = 13) infected groups. For each group, 3 birds and 10 birds were used for sampling and clinical observation, respectively. Chickens were inoculated via the eye drop/intranasal route at a dose of 10^5.0^ EID_50_/100 μL per bird, or with 100 μL PBS as the negative control. The birds were monitored for clinical signs daily for 10 dpi. Three birds were euthanized for gross lesion observation, and samples of the heart, liver, spleen, lung, kidney, brain, thymus, bursa of Fabricius, pancreas, and duodenum were collected to detect virus titration in DF-1 cells at 4 dpi. The virus titers were determined as the TCID_50_ per gram (log_10_TCID_50_/g tissue) using the endpoint method of Reed and Muench [47]. Other tissues, including spleen, thymus and bursa, lung, brain, and duodenum, were processed for the observation of pathological changes.

### 4.11. Statistical Analysis

All data were analyzed with GraphPad Prism software version 6.0 (GraphPad Software Inc., La Jolla, CA, USA). The *p*-values between the samples were generated using unpaired two-tailed Student’s *t*-test. All experiments were repeated at least three times and the results were presented as the mean ± standard deviation (SD). A *p*-value < 0.05 was considered statistically significant. *p*-values are indicated by asterisks (* *p* < 0.05, ** *p* < 0.01, *** *p* < 0.001).

## Figures and Tables

**Figure 1 ijms-24-00980-f001:**
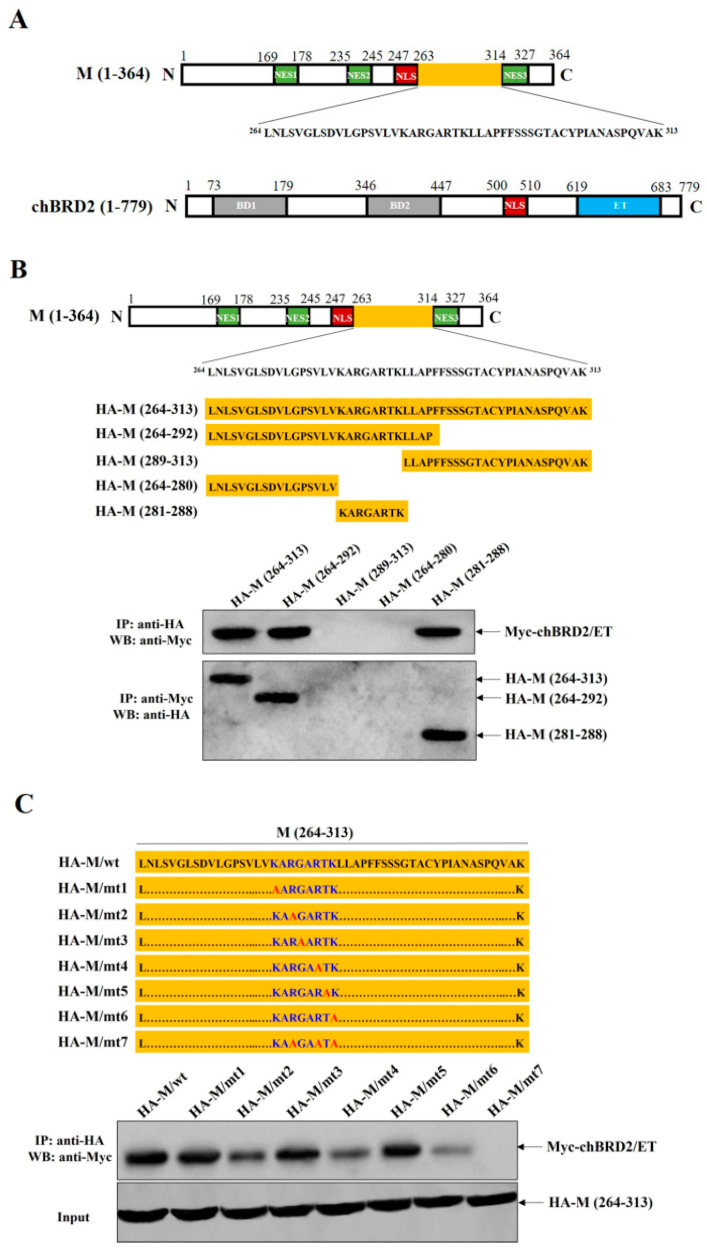
Basic residues R283, R286, and K288 of M protein are crucial for its interaction with chBRD2/ET. (**A**) Schematic diagram of NDV M protein and chBRD2. NLS, nuclear localization signal; NES, nuclear export signal; BD, bromodomain; ET, extra-terminal domain. (**B**) Characterization of the shortest amino acid region in the C-terminus (aa 264–313) of M protein interacting with chBRD2/ET. (**C**) Characterization of the key residues in the aa 281–288 of M protein that interacts with chBRD2/ET.

**Figure 2 ijms-24-00980-f002:**
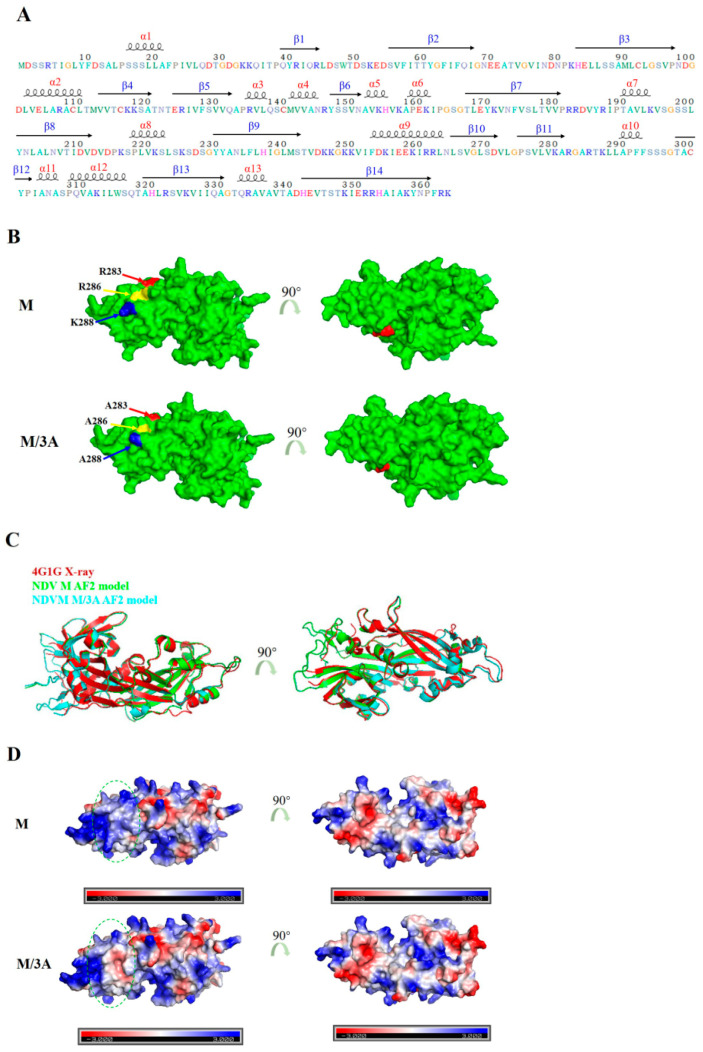
The M/3A protein has a changed electrostatic pattern. (**A**) Analysis of the secondary structure composition in the NDV M protein. (**B**) The 3D monomeric structural model of the M and M/3A proteins generated by AlphaFold2 (AF2) and PyMOL 2.5.4. Red, yellow, and blue represent the amino acids R283, R286, and K288, or A283, A286, and A288. (**C**) The cartoon drawing of the monomeric structure of M and M/3A proteins generated by PyMoL 2.5.4. (**D**) The electrostatic representations of the 3D monomeric structural model of the M and M/3A proteins generated by the APBS Electrostatics module of PyMoL 2.5.4. Blue indicates positively charged regions; red indicates negatively charged regions.

**Figure 3 ijms-24-00980-f003:**
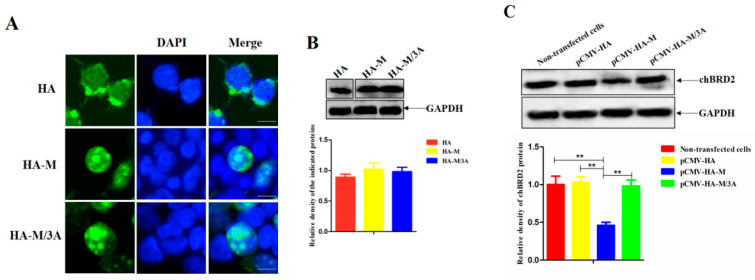
The M/3A protein abrogates the decreased expression of endogenic chBRD2. (**A**) The subcellular localization of M and M/3A proteins in plasmid-transfected cells. DAPI (blue) counterstained the nuclei. The original magnification was 1 × 100. Scale bars represent 10 μm. (**B**) Detection of the expression of HA tag, HA-M, and HA-M/3A proteins by Western blotting. (**C**) The effect of HA tag, HA-M, or HA-M/3A expression on endogenic chBRD2 expression. Asterisk denotes significant difference (**, *p* < 0.01).

**Figure 4 ijms-24-00980-f004:**
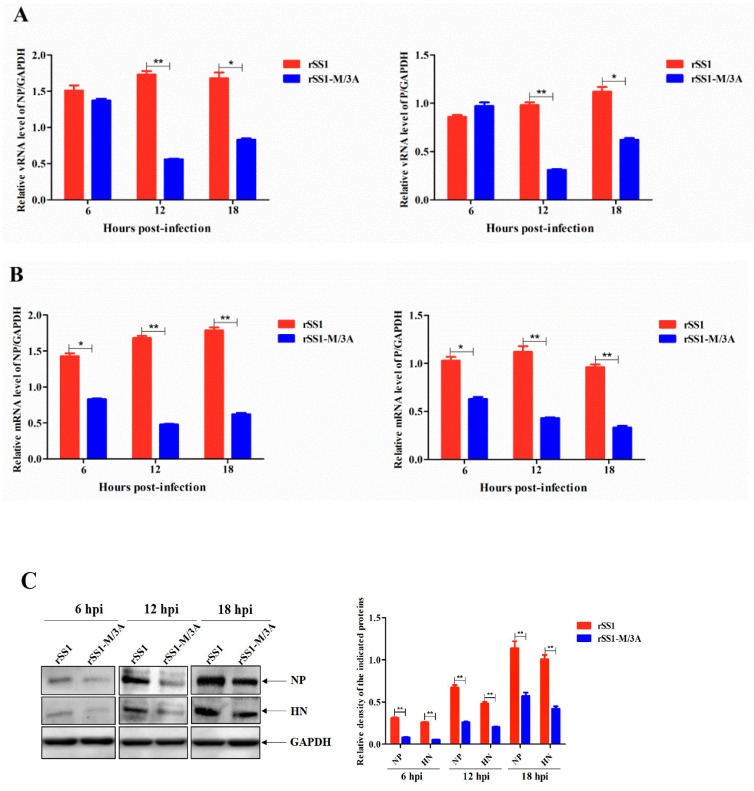
The M/3A protein causes the reduced viral RNA synthesis and transcription of recombinant NDV. (**A**) Viral RNA synthesis and transcription (**B**) corresponding to the *NP* and *P* genes in virus-infected cells were detected by qRT-PCR at different time points. (**C**) The expression of viral NP and HN proteins was detected by Western blotting. Asterisks denote significant differences (*, *p* < 0.05; **, *p* < 0.01).

**Figure 5 ijms-24-00980-f005:**
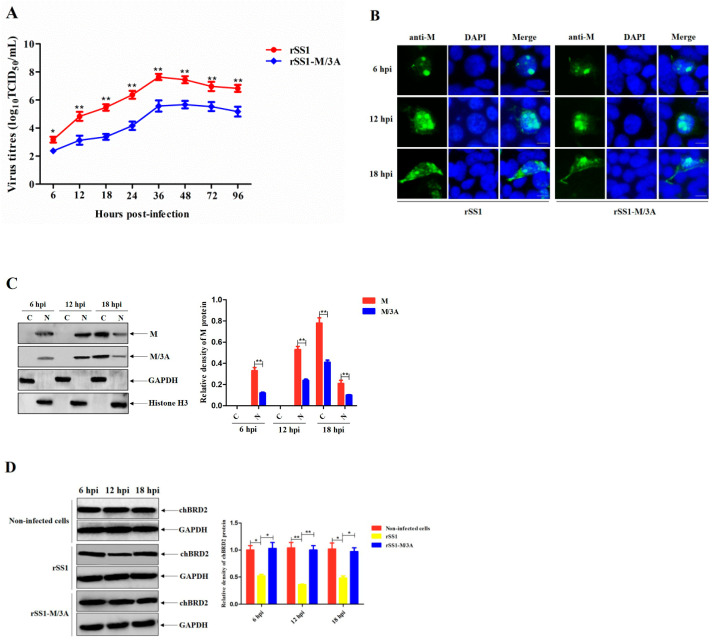
The M/3A protein leads to decreased replication ability of recombinant NDV. (**A**) The multicycle growth kinetics of rSS1 and rSS1-M/3A were determined in cells at different time points. (**B**) The subcellular localization of M and M/3A proteins in virus-infected cells was examined by IFA. DAPI (blue) counterstained the nuclei. The original magnification was 1 × 100. Scale bars represent 10 μm. (**C**) The cytoplasmic (C) or nuclear (N) distribution of M and M/3A proteins in virus-infected cells was detected by Western blotting. (**D**) The effect of rSS1 and rSS1-M/3A infection on the expression of endogenic chBRD2 at different time points. Asterisks denote significant differences (*, *p* < 0.05; **, *p* < 0.01).

**Figure 6 ijms-24-00980-f006:**
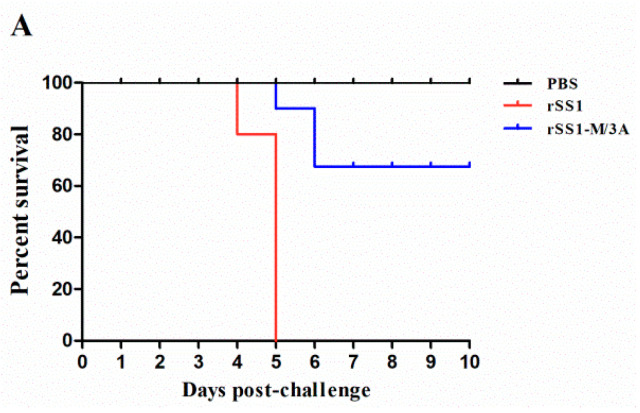
The M/3A protein attenuates the pathogenicity of recombinant NDV in SPF chickens. (**A**) Survival curves of 4-week-old SPF chickens inoculated with PBS, rSS1, or rSS1-M/3A. (**B**) Viral load in the collected tissues of 4-week-old chickens infected with the two viruses at 4 dpi. Asterisks denote significant differences (***, *p* < 0.001). (**C**) Histopathology of the tissue samples collected from rSS1-, rSS1-M/3A-, or PBS-inoculated 4-week-old chickens at 4 dpi. Empty arrow indicates the severe histopathological changes in the tissues of rSS1-infected chickens; black arrow indicates the slight histopathological changes in the tissues of rSS1-M/3A-infected chickens.

**Table 1 ijms-24-00980-t001:** Biological characteristics of the parental and mutant viruses *.

Virus	Pathogenicity		Virus Titer		
	MDT (h)	ICPI	EID_50_ mL^−1^	TCID_50_ mL^−1^	HA
rSS1	53 ± 2	1.90 ± 0.02	10^8.83^	10^8.71^	8 log_2_
rSS1-M/3A	102 ± 2	1.63 ± 0.01	10^6.37^	10^6.58^	5 log_2_

* MDT, mean death time; ICPI, intracerebral pathogenicity index; EID_50_, 50% egg infectious dose; TCID_50_, 50% tissue culture infective dose.

## Data Availability

All data supporting the findings of this research are available within the paper and within its Appendix A published online.

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
