# Peer review of "Mutation of Basic Residues R283, R286, and K288 in the Matrix Protein of Newcastle Disease Virus Attenuates Viral Replication and Pathogenicity"

_ijms, 2023, doi:10.3390/ijms24020980_

Round 1

Reviewer 1 Report

It is a well-written manuscript without typos. The experimental part seems to be fine, but the protein modelling part needs significant improvement. Based on literature search, the authors have been working with NDV for a long time and have certainly gained a lot of experimental experience. In this paper, they managed to identify three basic residues on the NDV M protein surface that play significant role in the protein-protein interaction with the chRBD2/ET. Moreover, they demonstrated that the M/3A mutant has nuclear localization, but the lack of chRBD2/ET interaction attenuates the virus.

I have generated 3D models for the WT and M/3A variants with the state-of-the-art AlphaFold2 (AF2) protein prediction method so as to compare with the described ones. Comparison of the AF2 models does not reveal a structural difference between the 47-55 aa region cited in the manuscript. I have attached the AF2 model files and a few supporting figures. I found a possible explanation why the Swiss-model used by the authors generated a short alpha helix in the region 47-55. This region is not resolved in the structure of the only available experimentally determined NDV M protein (PDB ID: 4G1G. https://www.rcsb.org/structure/4g1g). Probably, the Swiss-model homology modelling algorithm somehow tried to fold a short a-helix to this region for the 3A mutant. Please, have a look at the attached alignment and AF2 model comparisons figures. I suggest that the authors should write, instead of a change in the 3D structure, the change in the electrostatic pattern is responsible for the disruption of the protein-protein interaction. Moreover, there is no need to use the dimer form from the X-ray structure to show these changes. I would also suggest trying other protein structure prediction applications such as phyre2 (http://www.sbg.bio.ic.ac.uk/~phyre2/html/page.cgi?id=index) or I-TASSER (https://zhanggroup.org/I-TASSER/) and compare the results, but AF2 is probably the most reliable of these. I used the APBS Electrostatics module of the PyMol2.5.4 to generate the electrostatic representations for the NDV M proteins.

Eventually, after the aforementioned modifications the manuscript can be published in IJMS.

Author Response

Comments: I have generated 3D models for the WT and M/3A variants with the state-of-the-art AlphaFold2 (AF2) protein prediction method so as to compare with the described ones. Comparison of the AF2 models does not reveal a structural difference between the 47-55 aa region cited in the manuscript. I have attached the AF2 model files and a few supporting figures. I found a possible explanation why the Swiss-model used by the authors generated a short alpha helix in the region 47-55. This region is not resolved in the structure of the only available experimentally determined NDV M protein (PDB ID: 4G1G. https://www.rcsb.org/structure/4g1g). Probably, the Swiss-model homology modelling algorithm somehow tried to fold a short a-helix to this region for the 3A mutant. Please, have a look at the attached alignment and AF2 model comparisons figures. I suggest that the authors should write, instead of a change in the 3D structure, the change in the electrostatic pattern is responsible for the disruption of the protein-protein interaction. Moreover, there is no need to use the dimer form from the X-ray structure to show these changes. I would also suggest trying other protein structure prediction applications such as phyre2 (http://www.sbg.bio.ic.ac.uk/~phyre2/html/page.cgi?id=index) or I-TASSER (https://zhanggroup.org/I-TASSER/) and compare the results, but AF2 is probably the most reliable of these. I used the APBS Electrostatics module of the PyMol2.5.4 to generate the electrostatic representations for the NDV M proteins.

Response: Thank you very much for your valuable and helpful suggestions. We agree with the reviewer’s comments that AlphaFold2 (AF2) is probably the most reliable prediction method for protein structure prediction. According to the reviewer’s suggestions, we have revised the 3D models and the secondary structure in the cartoon drawing of the monomeric structure for the M and M/3A proteins. Meanwhile, the APBS Electrostatics module of the PyMol2.5.4 to generate the electrostatic representations for the M and M/3A proteins was also resubmitted. In addition, “the change in the electrostatic pattern is responsible for the disruption of the protein-protein interaction” has also been corrected in our revised manuscript. Please see Figure 2B, 2C and 2D, and lines 18-19, 77-78, 117-124, 291-297 (marked in red).

Thank you again for your helpful comments and suggestions.

Reviewer 2 Report

Authors show that basic amino acids R283, R286 and K288 of the NDV Matrix protein abrogate the interaction with ChBRD 2 and therefore can be considered as NDV virulence factors. Paper is well described in particular introduction and results. The discussion however could be improved considerably for most because the first two paragraphs are redundant to  or should have been mentioned in the introduction. 

Other remarks:

-the construction of some plasmids are not described in M&M; for instance  for construction of  pCMV-HA the reader is referred to ref 17. However in that paper it appears that pCMV-HA was commercially purchased. It would be helpful if at least the abbreviation HA is explained as influenza haemagglutinin.

The subsection sequence within the results section is questionable: 2.4 is more related to 2.6 than to 2.5 which in turn is more related to section 2.3.

Author Response

Comments: Authors show that basic amino acids R283, R286 and K288 of the NDV Matrix protein abrogate the interaction with ChBRD2 and therefore can be considered as NDV virulence factors. Paper is well described in particular introduction and results. The discussion however could be improved considerably for most because the first two paragraphs are redundant to or should have been mentioned in the introduction.

Response: Thanks for your helpful and valuable comments. We agree with the reviewer’s comments. The first two paragraphs in discussion section have been improved to be associated with the subsequent discussion. Meanwhile, some contents have been mentioned in the introduction. Please see lines 46-49, 271-291 (marked in blue).

Comments: the construction of some plasmids are not described in M&M; for instance, for construction of pCMV-HA the reader is referred to ref 17. However, in that paper it appears that pCMV-HA was commercially purchased. It would be helpful if at least the abbreviation HA is explained as influenza haemagglutinin.

Response: Thanks for your helpful and valuable comments. The recombinant plasmids pCMV-HA-M, pCMV-HA-M(264-313), pCMV-Myc-chBRD2/ET has been generated in our previous studies, so the construction of these plasmids was not described. But the construction of other plasmids was described in the manuscript in M&M (4.3. Plasmid construction). Meanwhile, the primer information for the construction of recombinant plasmids was provided in Supplementary materials (Table S1). In addition, the abbreviation of HA and the origin of pCMV-HA were described in the revised manuscript. Please see lines 355-356 (marked in blue).

Comments: The subsection sequence within the results section is questionable: 2.4 is more related to 2.6 than to 2.5 which in turn is more related to section 2.3.

Response: Thanks for your helpful and valuable comments. We agree with the reviewer’s comments. According to the reviewer’s suggestion, we have adjusted the order of 2.3 to 2.6.

Thank you again for your helpful comments and suggestions.

Round 2

Reviewer 1 Report

I would like to clarify a few things about the protein modelling. The authors wrote this in line 373-375: „In addition, the amino acid sequences of M and M/3A proteins were uploaded to the AlphaFold Protein Structure Database (https://alphafold.ebi.ac.uk/) to generate 3D models.”

I see that the authors are not yet experienced in the use of AF. You can't run AF via the (https://alphafold.ebi.ac.uk/) site, only the pregenerated structures can be downloaded. Basically, there are two ways to run AF: one is to install AF on your own GPU image from the AF github page or use google colab.

https://alphafold.ebi.ac.uk/faq Look for the question: “What if I can’t find the protein I’m interested in?”

https://github.com/deepmind/alphafold/

https://colab.research.google.com/github/deepmind/alphafold/blob/main/notebooks/AlphaFold.ipynb

I know that they used my previously shared pdb files. Therefore, I suggest that to write simply „the protein models were generated by AF” or something like that.

Don’t need to write this sentence in line 375: “The structural models of the M and M/3A proteins were constructed based on the reported crystal structure of NDV M protein (4G1G).”

Because, AF used its whole database to gather information for 3D structure generation. Instead of this sentence the authors should write that the 4G1G (https://www.rcsb.org/structure/4g1g, Newcastle disease virus (STRAIN AUSTRALIA-VICTORIA/32) is the solely structural template.

https://blast.ncbi.nlm.nih.gov/Blast.cgi?CMD=Get&RID=U9SYZ4CG013

Author Response

Comment: I would like to clarify a few things about the protein modelling. The authors wrote this in line 373-375: „In addition, the amino acid sequences of M and M/3A proteins were uploaded to the AlphaFold Protein Structure Database (https://alphafold.ebi.ac.uk/) to generate 3D models.”

I see that the authors are not yet experienced in the use of AF. You can't run AF via the (https://alphafold.ebi.ac.uk/) site, only the pregenerated structures can be downloaded. Basically, there are two ways to run AF: one is to install AF on your own GPU image from the AF github page or use google colab.

https://alphafold.ebi.ac.uk/faq Look for the question: “What if I can’t find the protein I’m interested in?”

https://github.com/deepmind/alphafold/

https://colab.research.google.com/github/deepmind/alphafold/blob/main/notebooks/AlphaFold.ipynb

I know that they used my previously shared pdb files. Therefore, I suggest that to write simply „the protein models were generated by AF” or something like that.

Don’t need to write this sentence in line 375: “The structural models of the M and M/3A proteins were constructed based on the reported crystal structure of NDV M protein (4G1G).”

Because, AF used its whole database to gather information for 3D structure generation. Instead of this sentence the authors should write that the 4G1G (https://www.rcsb.org/structure/4g1g, Newcastle disease virus (STRAIN AUSTRALIA-VICTORIA/32) is the solely structural template.

https://blast.ncbi.nlm.nih.gov/Blast.cgi?CMD=Get&RID=U9SYZ4CG013

Response: Thanks for your valuable comments and suggestions. As the reviewer said,  we directly used the reviewer's previously shared pdb files. Therefore, according to the reviewer's suggestions, we corrected the corresponding sentences as follows "In addition, the protein structural models of M and M/3A were generated by AlphaFold. The 4G1G (https://www.rcsb.org/structure/4g1g), crystal structure of Newcastle disease virus (STRAIN AUSTRALIA-VICTORIA/32) M protein, is the solely structural template." Please see the revised manuscript, marked in red.